# Evaluating HIV-1 Infectivity and Virion Maturation across Varied Producer Cells with a Novel FRET-Based Detection and Quantification Assay

**DOI:** 10.3390/ijms25126396

**Published:** 2024-06-10

**Authors:** Aidan McGraw, Grace Hillmer, Jeongpill Choi, Kedhar Narayan, Stefania M. Mehedincu, Dacia Marquez, Hasset Tibebe, Kathleen L. DeCicco-Skinner, Taisuke Izumi

**Affiliations:** 1Department of Biology, College of Arts and Sciences, American University, Washington, DC 20016, USA; amcgraw@american.edu (A.M.); gh2297a@american.edu (G.H.); jc9126a@american.edu (J.C.); kn0119a@american.edu (K.N.); sm4872a@american.edu (S.M.M.); dm6732a@american.edu (D.M.); ht8146a@american.edu (H.T.); decicco@american.edu (K.L.D.-S.); 2District of Columbia Center for AIDS Research, Washington, DC 20052, USA

**Keywords:** human immunodeficiency virus type I (HIV-1), forster resonance energy transfer (FRET), virion maturation, comparative viral analysis, fluorescent microscopic imaging

## Abstract

The maturation of HIV-1 virions is a crucial process in viral replication. Although T-cells are a primary source of virus production, much of our understanding of virion maturation comes from studies using the HEK293T human embryonic kidney cell line. Notably, there is a lack of comparative analyses between T-cells and HEK293T cells in terms of virion maturation efficiency in existing literature. We previously developed an advanced virion visualization system based on the FRET principle, enabling the effective distinction between immature and mature virions via fluorescence microscopy. In this study, we utilized pseudotyped, single-round infectious viruses tagged with FRET labels (HIV-1 Gag-iFRET∆Env) derived from Jurkat (a human T-lymphocyte cell line) and HEK293T cells to evaluate their virion maturation rates. HEK293T-derived virions demonstrated a maturity rate of 81.79%, consistent with other studies and our previous findings. However, virions originating from Jurkat cells demonstrated a significantly reduced maturation rate of 68.67% (*p* < 0.0001). Correspondingly, viruses produced from Jurkat cells exhibited significantly reduced infectivity compared to those derived from HEK293T cells, with the relative infectivity measured at 65.3%. This finding is consistent with the observed relative maturation rate of viruses produced by Jurkat cells. These findings suggest that initiation of virion maturation directly correlates with viral infectivity. Our observation highlights the dynamic nature of virus–host interactions and their implications for virion production and infectivity.

## 1. Introduction

Human immunodeficiency virus type 1 (HIV-1) continues to pose a significant global health challenge. HIV-1 targets CD4+ T-cells by binding its gp120 glycoprotein to the CD4 receptor and co-receptors (CCR5 or CXCR4), followed by fusion with the host cell membrane [1,2,3]. Upon membrane fusion, the viral core is released into the cytoplasm, where reverse transcription converts viral RNA into DNA via the viral reverse transcriptase [4,5,6]. The synthesized viral DNA is then transported into the nucleus and integrates into the host genome through viral integrase [7,8,9,10,11]. The integrated viral DNA, called a provirus, utilizes the host cell machinery to transcribe and translate viral RNA and component proteins, respectively, which are assembled at the plasma membrane to produce immature viral particles [12,13,14]. Finally, these particles are released from the host cell and mature into infectious viruses through the viral protease cleavage of Gag polyproteins [15,16,17]. The advancement of combination antivirals targeting different stages of the HIV life cycle including entry inhibitors (fusion inhibitors and CCR5 antagonists), reverse transcriptase inhibitors (NRTIs and NNRTIs), integrase inhibitors, protease inhibitors, and capsid inhibitors has dramatically improved the management of HIV-1 [18,19,20,21,22,23]. These inhibitors act by blocking specific processes, thereby preventing the virus from successfully replicating and spreading within the host. Specifically, the development of capsid inhibitors has emerged as a promising strategy for long-acting antiviral treatments against HIV-1 [24,25,26,27,28,29]. These novel inhibitors, targeting critical steps in both the early and late stages of the virus lifecycle, notably disrupt the formation of infectious particles. This innovation underscores the need for a comprehensive understanding of the virion maturation process, a key phase in the HIV-1 life cycle where non-infectious particles are transformed into infectious virions.

Our previous research developed a unique tool for detecting virus maturation utilizing a fluorescence resonance energy transfer (FRET)-based technique (Figure 1A) [30]. This approach enables the semi-automatic evaluation of virion maturation through fluorescent microscopy, effectively minimizing human bias. This novel tool corresponded well to the virion maturation ratio confirmed by the Transmission Electron Microscopy (TEM) assay [30]. Therefore, we can more easily address virion maturation efficiency in multiple virus samples using our tool, which offers crucial insights into the structural changes that are vital for virus infectivity. Building upon this, our current study explores virion maturation in different cellular environments. Although T-cells are recognized as the primary source for HIV-1 replication [5,31,32,33,34,35,36], most existing research assessing viral maturation predominantly utilizes the HEK293T human embryonic kidney cell line [37,38,39,40]. Utilizing the FRET-based system, we generated infectious FRET-labeled viruses in both HEK293T and Jurkat cells, an immortal human T-cell leukemia cell line. Surprisingly, we observed that viruses produced by Jurkat cells exhibited a significantly lower maturation efficiency (approximately 69%) compared to those from HEK293T cells (approximately 82%). In alignment with the observed differences in maturation rates, the infectivity of viruses from Jurkat cells was approximately half that observed in viruses from HEK293T cells. Given that CD4+ T-cells are the principal producers of HIV-1, these cells have evolved multiple antiretroviral mechanisms. For example, the HIV-1 viral infectivity factor (Vif) is a crucial accessory molecule for HIV-1 replication in vivo. However, it is not necessary for HEK293T cells, as these kidney-derived cells do not express the potent antiviral host factor APOBEC3G [30,41,42]. It is highly plausible that kidney cells, having not been exposed to lentiviruses throughout evolutionary history, exhibit greater susceptibility to these viruses compared to T-cells, which have been concurrently evolving to resist viral infections. This study is at the forefront of comparing virion maturation and infectivity among cell lines from different origins, highlighting a novel approach in this field of HIV-1 research.

## 2. Results

### 2.1. Virion Maturation Efficiency in Different Producer Cells

In this study, we quantified the proportion of mature and immature virions in FRET-labeled viruses produced by HEK293T (kidney) and Jurkat (T-cell leukemia) cell lines. In each sample, we captured 21 images, from which we created binary images based on the emitted and excited images of Yellow Fluorescent Protein (YFP) (Figure 1B). We then calculated the FRET ratio for each virion by dividing the intensity of YFP (the FRET acceptor) by the total Cyan Fluorescent Protein (CFP) intensity, which serves as the FRET donor. After calculating the FRET ratio for each virion, based on the extracted signal intensities of the FRET donor and acceptor, we applied kernel density estimation to the histograms of FRET efficiencies. The area where the HIV-1 Gag-iFRET∆Env curve overlapped with the HIV-1 Gag-iFRET∆PR∆Env curve was identified as the proportion of immature virions, as per our established methodology (Figure 1C) [30]. In our analysis, we examined 14,021 and 22,779 particles, as well as 20,457 and 23,181 particles of HIV-1 Gag-iFRET∆Env- and HIV-1 Gag-iFRET∆PR∆Env-labeled virions produced by HEK293T and Jurkat cells, respectively, across three independent experiments. The mature virion proportion in the HIV-1 Gag-iFRET∆Env population from HEK293T cells was 81.79% ± 0.06 (Figure 1D), consistent with our previous findings and electron microscopic counts reported in other studies [30,37,38,39,40]. In contrast, Jurkat cells produced mature virions at a lower rate of 68.67% ± 0.04 (Figure 1D), which is significantly different from that of HEK293T cells (*p*-value < 0.0001). Furthermore, we quantified the mean fluorescent intensity (MFI) of the YFP signal in both mature and immature virions. The results confirmed that the average incorporation of fluorescent proteins, indicated by the MFI of YFP (Figure 1E), was similar in both mature and immature virions from HEK293T and Jurkat cells (23,870.96 ± 2410.36 vs. 21,705.86 ± 278.20 in mature virions and 30,841.07 ± 5596.51 vs. 26,560.47 ± 2385.33 in immature virions, respectively, and 24,091 ± 2857.71 vs. 24,314.76 ± 2046.71 in ΔPR). These findings indicate that the number of incorporated fluorescent proteins is comparable, ensuring that the FRET energy transfer efficiency is neither under nor overestimated in virions produced by either HEK293T or Jurkat cells.

### 2.2. Virus Infectivity in Different Producer Cells

To further examine the characteristic differences between viruses produced by HEK293T and Jurkat cells, we performed a single-round infection assay using TZM-bl cells to assess the infectivity of FRET-labeled viruses originating from these cell lines. The infectivity of viruses from Jurkat cells was significantly lower compared to those from HEK293T cells, measured at 67.38% ± 11.2% compared to 109.31% ± 17.91%, respectively (Figure 2). This discrepancy is consistent with the observed differences in the maturation rates of viruses produced by Jurkat cells. (Figure 1).

## 3. Discussion

The findings of this study provide groundbreaking insights into the maturation process of HIV-1 virions and their impact on viral infectivity, challenging several established concepts in the field. Our utilization of an advanced FRET-based visualization system has allowed for a detailed comparative analysis of virion maturation in two distinct cellular environments: HEK293T and Jurkat cell lines. One of the most notable findings of our study is that virus maturation and subsequent infectivity in T-cell lines, which are derived from the natural targets for HIV-1, are lower compared to kidney-derived cell lines. HEK293T cells exhibited a reduced rate of immature virions (18.21%), whereas virions derived from Jurkat cells displayed a higher rate of immaturity (31.33%), correlating with its respective infectivity (relative to 61.76% ± 4.81% for Jurkat cell-derived virions compared to the infectivity observed in virions from HEK293T cells, which was set to 100%). Recent studies have provided deeper insights into the role of inositol hexakisphosphate (IP6) in HIV-1 capsid assembly, which is a critical factor in virus infectivity [43]. These studies reveal that the immature Gag lattice of HIV-1 virions enriches IP6, facilitating capsid maturation. This process is characterized by proteolysis of the Gag polyprotein by the viral protease, which induces vital conformational changes in the capsid protein (CA). These changes led to the formation of a conical, cone-shaped mature capsid consisting of over 1000 CA copies and forming approximately 200 hexamers and 12 pentamers, collectively known as capsomers. Sowd et al. showed that IP6-depleted T-cell lines produce virion particles with incompletely cleaved Gag proteins [44]. This study indicated that virions from IP6-deficient cells exhibited lower quantities of mature MA and CA cleavage products and higher proportions of MA-CA-SP1/MA-CA cleavage intermediates than virions from wild-type cells. In the context of our FRET-based assay, which specifically detects cleavage of the Gag polyprotein between MA-CA, our FRET detection system can potentially monitor the IP6-dependent efficiency of Gag polyprotein cleavage. Furthermore, they characterized virus particle morphology using electron microscopy and calculated that mature particles in another T-cell leukemia cell line, MT-4 cells, comprised 62% of the total virions, which aligns with the results from our FRET detection system in Jurkat cells (Figure 1C [II] and D). Given that the majority of inositol synthesis in humans occurs in kidney tissue [45], it is reasonable to infer that HEK293T cells would exhibit abundant IP6 expression. This is likely to contribute to enhanced virion maturation efficiency. It has also been suggested that IP6 plays a crucial role in facilitating the assembly of the immature HIV-1 Gag lattice [46]. Consequently, the elevated expression of IP6 in HEK293T cells might enhance the assembly of Gag precursors at the budding site prior to virion release, potentially initiating the trimerization of the MA domain. This, in turn, could enhance the efficient incorporation of Env proteins into progeny virions [47,48]. We need further molecular biological assays to confirm these hypotheses. However, since IP6 is a metabolite, it is not easy to measure its levels in virions and cells. We are currently developing a mass-spectrometry-based assay to determine the amount of IP6 in virion produced by Jurkat and HEK293T cells.

Another study employing a similar concept of FRET-based assay with different FRET pair proteins revealed that the activation of HIV-1 viral protease occurs during assembly and budding before the release of particles in HEK293T cells [49]. The early maturation observed in HEK293T cells could lead to an increase in progeny virion maturation. This effect likely accounts for the elevated maturation ratio observed in HEK293T cells in our study (Figure 1). This early maturation could also result in enhanced viral infectivity compared to virions originating from Jurkat cells (Figure 2).

Several accessory molecules, such as Vif, are not required to produce infectious viruses from HEK293T cells [30,41,42]. This is due to the absence of antiviral host factors in HEK293T cells that would normally be counteracted by viral accessory proteins. This discrepancy may hint at the heightened susceptibility of kidney cells to retrovirus infection, potentially attributable to their lack of prior exposure to these viral components. Thus, the environment in HEK293T cells is optimized not only by the absence of antiviral host protein expression but also by the enrichment of host cofactors, which collectively contribute to increased virus infectivity.

Our research addresses a critical gap in the existing literature by presenting a comparative analysis of the efficiency of Gag polyprotein cleavage in virions produced by kidney cells (HEK293T) and T-cells (Jurkat). This comparison is vital, considering T-cells are the primary source of HIV-1 replication in vivo. The observed differences in maturation efficiencies and infectivity rates between HEK293T and Jurkat cells highlight the importance of considering the cellular context in HIV-1 biology and pathogenesis studies. Furthermore, our findings have significant implications for the evaluation of antiviral effects, particularly in the context of long-acting treatments with maturation and capsid inhibitors. These inhibitors target various stages of virus infection, including viral budding and virion maturation [15,47,50]. The substantial disparities in viral maturation rates between Jurkat cells and HEK293T cells, the latter of which is a standard cellular resource for in vitro virus production, suggest that relying solely on HEK293T cells could lead to misleading assessments of antiviral efficacy. Therefore, our study indicates the necessity of including T-cell-based assessments to more accurately determine the antiviral efficacy of maturation and capsid inhibitors in virus maturation.

In conclusion, the results of our study not only enhance our understanding of HIV-1 pathogenesis but also emphasize the complexity of the virus life cycle. The unique insights gained into the relationship between virion maturation and infectivity open new pathways for future research, potentially leading to novel therapeutic approaches in the fight against HIV-1.

## 4. Materials and Methods

### 4.1. Plasmid DNA Construction

The plasmid DNA utilized for HIV-1 Gag-iFRET labeling virus production incorporates an efficient intramolecular FRET pair, ECFP∆C11 and cp173Venus (Figure 1A) [51]. These genes are positioned between the MA and CA regions of the HIV-1 Gag protein based on the HIV-1 Gag-iGFP construct [52]. The FRET pair genes, along with the junction with the MA and CA regions, are flanked by HIV-1 protease cleavage sites (SQNYPIVQ). This design enables the HIV-1 protease to specifically cleave at these sites, resulting in a modification of the FRET signal during the maturation of the virion. Additional details are provided in our previous publication [30].

### 4.2. Virus Production and Cell Culture

HIV-1 Gag-iFRET plasmid DNA was transfected into HEK293T cells, kindly provided by Dr. Zachary Klase at Drexel University, using a PEI transfection method (Polysciences, Warrington, PA, USA, https://www.polysciences.com/default/catalog-products/polymers/polyethylenimine-pei) and into Jurkat cells, kindly provided by Dr. Eric Free at NC-Frederick, using lipofectamine LTX transfection reagent (Invitrogen, Waltham, MA, USA, https://www.thermofisher.com/order/catalog/product/15338100?SID=srch-srp-15338100). HEK293T and TZM-bl cells, which were obtained through the NIH HIV Reagent Program (currently BEI Resources), Division of AIDS, NIAID, (Manassas, VA, USA) [53,54,55,56,57,58] were cultured in Dulbecco’s modified Eagle’s medium (Cytiva, Marlborough, MA, USA, https://www.cytivalifesciences.com/en/us/shop/cell-culture-and-fermentation/media-and-feeds/classical-media/hyclone-dulbecco’s-modified-eagle-medium-dmem-with-high-glucose-liquid-p-05890) supplemented with 10% fetal bovine serum (Gibco, USA, https://www.thermofisher.com/order/catalog/product/10082147), 1% penicillin–streptomycin–glutamine (Gibco, USA, https://www.thermofisher.com/order/catalog/product/10378016?SID=srch-srp-10378016 ), and 1% GlutaMax (Gibco, USA, https://www.thermofisher.com/order/catalog/product/35050061?SID=srch-srp-35050061) (D10) at 37 °C in a 5% CO_2_ environment. Suspended Jurkat cells were cultured in RPMI 1640 media (Cytiva, USA, https://www.cytivalifesciences.com/en/us/shop/cell-culture-and-fermentation/media-and-feeds/classical-media/hyclone-rpmi-1640-media-liquid-p-05892) with the same supplements (R10). To produce FRET-labelled virions, HEK293T cells (7.0 × 10^6^ cells/10 cm dish) and Jurkat cells (10 × 10^6^ cells/10 cm dish) were co-transfected with pHIV-1 Gag-iFRETΔEnv or iFRETΔPRΔEnv, along with the parental plasmids pNL4-3ΔEnv or pNL43ΔPRΔEnv, and the pSVIII-92HT593.1 dual-tropic HIV-1 Env expression plasmid, kindly provided by Dr. Viviana Simon at Mount Sinai, using a transfection ratio of 1:10:2.5. The culture medium was replaced with fresh D10 medium after 3.5 h for PEI transfection. Virus-containing supernatants were collected 48 h post-transfection. The supernatants were then filtered through a 0.45 µm sterile polyvinylidene difluoride (PVDF, CELLTREAT, USA, https://www.celltreat.com/product/229744/) membrane and concentrated up to 10-fold using Lenti-X Concentrator (TaKaRa, https://www.takarabio.com/products/gene-function/viral-transduction/lentivirus/lentivirus-concentration). The virus pellet was resuspended in 100 µL of phosphate-buffered saline (PBS) without phenol red (Cytiva, USA, https://www.cytivalifesciences.com/en/us/shop/cell-culture-and-fermentation/buffers-and-process-liquids/balanced-salt-solutions/hyclone-phosphate-buffered-saline-solution-p-00520).

### 4.3. FRET-Based Virion Visualization

To visualize HIV-1 Gag-iFRETΔEnv/iFRETΔPRΔEnv-labeled virions, we diluted the concentrated virus supernatant 800-fold in 0.22 µm PVDF-filtered Hank’s Balanced Salt Solution (HBSS) without Phenol Red (Cytiva, USA, https://www.cytivalifesciences.com/en/us/shop/cell-culture-and-fermentation/buffers-and-process-liquids/balanced-salt-solutions/hyclone-hank’s-1x-balanced-salt-solutions-p-00537) and load 350 µL onto a glass-bottomed 8-chamber slide (ibidi, USA, https://ibidi.com/chambered-coverslips/13--slide-8-well-ibitreat.html). The slide was then incubated overnight at 4 °C. Single-virion images were captured using an A1R MP+ Multiphoton Confocal Microscope (Nikon, https://www.microscope.healthcare.nikon.com/products/confocal-microscopes/a1r-hd-mp). Two sets of 21 images were automatically acquired for each sample under optimal focus conditions. The first set employed a 457.9 nm laser for CFP excitation, capturing emissions through 482 nm/35 nm and 540 nm/30 nm filter cubes for CFP and YFP signals, respectively (FRET images). The second set used a 514.5 nm laser for Venus excitation, with emissions read through a 540 nm/30 nm filter cube to detect the YFP signal. The maturation status was quantified as FRET efficiency relative to signals detected in HIV-1 Gag-iFRET∆PRΔEnv-labeled virions. Images were captured in RAW ND2 format and converted to TIFF files using NIS-Elements Viewer, a free standalone program to view image files and datasets. Binary images, generated based on the YFP signal, provided XY coordinates for each particle. The centroid of the XY position for each virion particle was determined, and a circle with a two-pixel radius centered on the calculated centroid was drawn to represent the virion map. Using this virion map, the FRET signal intensity for each virion was extracted from the raw data, and the FRET ratio (YFP/CFP) was calculated for each particle. Additionally, the MFI of YFP within the circle of the virion map was also extracted from the raw data. Histograms displaying the distribution of FRET ratio values were generated, with 100 bins division, and kernel density estimation curves were plotted alongside. The proportion of the total kernel density estimation area overlapping with the HIV-1 Gag-iFRET∆PRΔEnv area was used to determine the proportion of immature virions. Image data analysis and subsequent calculations of mature and immature virion proportions were performed using an updated in-house MATLAB program [30].

### 4.4. Infectivity Assays

Viral titers were determined using the HIV-1 Gag p24 DuoSet ELISA (R&D Systems, USA, https://www.rndsystems.com/products/hiv-1-gag-p24-duoset-elisa_dy7360-05). After seeding TZM-bl cells at 1 × 10^4^ in flat-bottom 96-well plates (CELLTREAT, USA, https://www.celltreat.com/product/229196/), they were exposed to an equivalent amount of virus (5 ng of HIV-1 p24 total) on the following day. The cells were incubated at 37 °C for 48 h in a CO_2_ incubator. Subsequently, luciferase activity in the infected cells was measured with the Luciferase Assay System (Promega, USA, https://www.promega.com/products/luciferase-assays/reporter-assays/luciferase-assay-system/?catNum=E1500) using a Molecular Devices SpectraMax Microplate Reader (Molecular Devices, https://info.moleculardevices.com/br/one-spectramax-endless-discoveries?cmp=701Rn0000076EtvIAE&utm_adgroup=NA-MPR-Brand-Spectramax-i-Series&utm_campaignid=932755352&utm_adgroupid=53348384368&utm_location=9007538&utm_keyword=molecular%20devices%20spectramax&utm_device=c&utm_placement=&utm_adpostion=&utm_network=g&utm_campaign=MD-GLO-BR-CC-PPC-Google-20240208-InfiniteDiscoveriesMMR&utm_medium=cpc&utm_source=adwords&utm_term=molecular%20devices%20spectramax&hsa_acc=4341129614&hsa_cam=932755352&hsa_grp=53348384368&hsa_ad=602940400491&hsa_src=g&hsa_tgt=aud-1110859176626:kwd-330068804608&hsa_kw=molecular%20devices%20spectramax&hsa_mt=p&hsa_net=adwords&hsa_ver=3&gad_source=1).

### 4.5. Statistical Analysis

Data were analyzed using standard statistical methods. The significance of differences in virion maturation and infectivity between the two cell types was determined using the Mann–Whitney U test with a *p*-value of less than 0.05, which is considered statistically significant. The statistical differences in YFP MFI between the two cell types were calculated using the Wilcoxon matched-pairs signed-rank test.

## Figures and Tables

**Figure 1 ijms-25-06396-f001:**
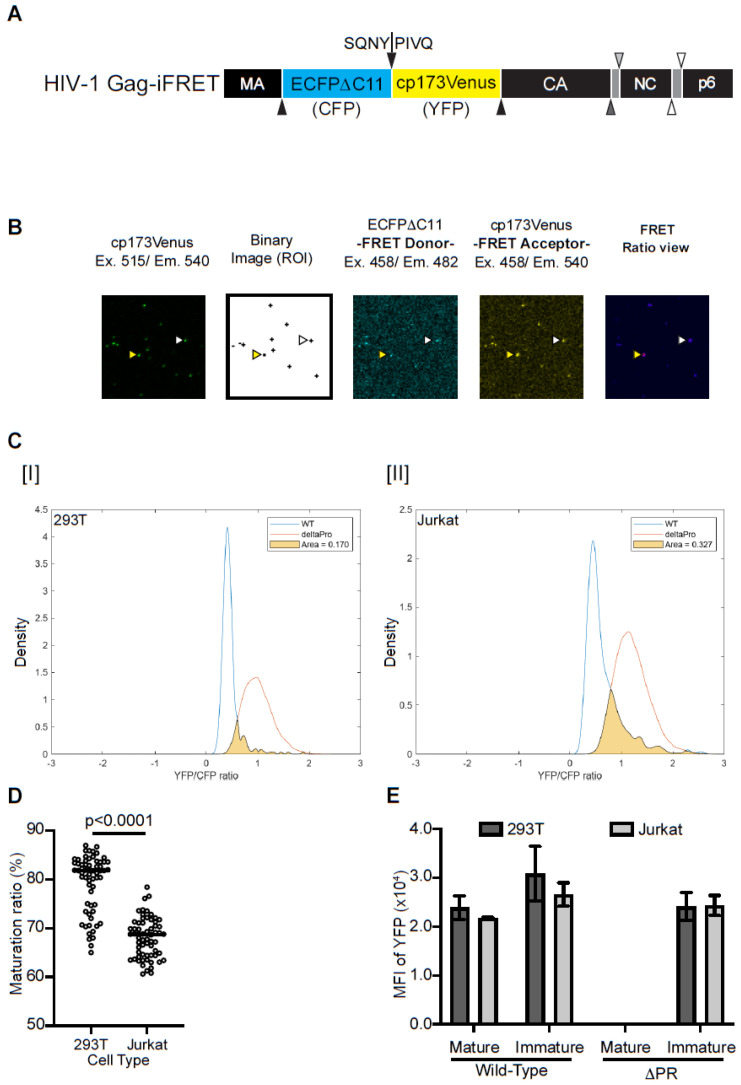
Quantitative analysis of mature virions from HEK293T and Jurkat cells. (**A**) Schematic representation of the HIV-1 Gag iFRET construct in the Gag region. ECFP∆C11 (CFP) and cp173Venus (YFP) are used for efficient single-molecule CFP and YFP FRET pairing. Arrowheads indicate the HIV-1 protease cleavage sites. CFP and YFP are connected with the protease cleavage site originally derived from the MA and CA junction. (**B**) Representative FRET-labeled HIV-1 Gag-iFRET∆Env virion images from Jurkat cells displayed in a single field. The left panel shows an image captured through the YFP excitation (515 nm) and emission (540 nm) channels (YFP channel). The second image shows the binary image, derived from the YFP channel, identifying virus particle locations as regions of interest (ROI). The third and fourth images are captured using CFP (FRET donor) excitation (454 nm), with CFP emission (483 nm, third) and YFP emission (540 nm, fourth) channels. The right panel shows the FRET efficiency ratio view, computationally constructed from FRET donor (CFP excitation/CFP emission) and acceptor (CFP excitation/YFP emission) images. The color spectrum indicates FRET signal strength, with high FRET appearing in red (yellow arrows) and decreasing FRET shifting toward blue (white arrows). (**C**) Kernel density estimation curves of FRET intensity in HIV-1 Gag-iFRET∆Env virions produced by HEK293T and Jurkat cells. The curves depict the distribution histograms of FRET intensity for virions labeled with HIV-1 Gag-iFRET∆Env from (I) HEK293T and (II) Jurkat cells. The *x*-axis represents the range of FRET intensity. The *y*-axis shows the adjusted density of curves for both HIV-1 Gag-iFRET∆Env and HIV-1 Gag-iFRET∆PR∆Env virions. The area under the curve for HIV-1 Gag-iFRET∆Env overlapping with that of HIV-1 Gag-iFRET∆PR∆Env is calculated and considered indicative of the proportion of immature virions within the total population of HIV-1 Gag-iFRET∆Env virions. (**D**) A scatter plot of mature virion quantification displays the median percentage of mature virions across 21 images derived from three independent experiments, culminating in a total of 63 data points. Statistical significance was assessed using the Mann–Whitney U test. (**E**) Comparison of MFI of YFP signals in mature and immature HIV-1 Gag-iFRET∆Env labeling virions. The levels of fluorescent protein incorporation, as indicated by the YFP signal, were not significantly different between mature and immature virions in either the wild-type or ∆PR viruses. The error bars represent the standard error from three independent experiments. Statistical significance was assessed using the Wilcoxon matched-pairs signed rank test.

**Figure 2 ijms-25-06396-f002:**
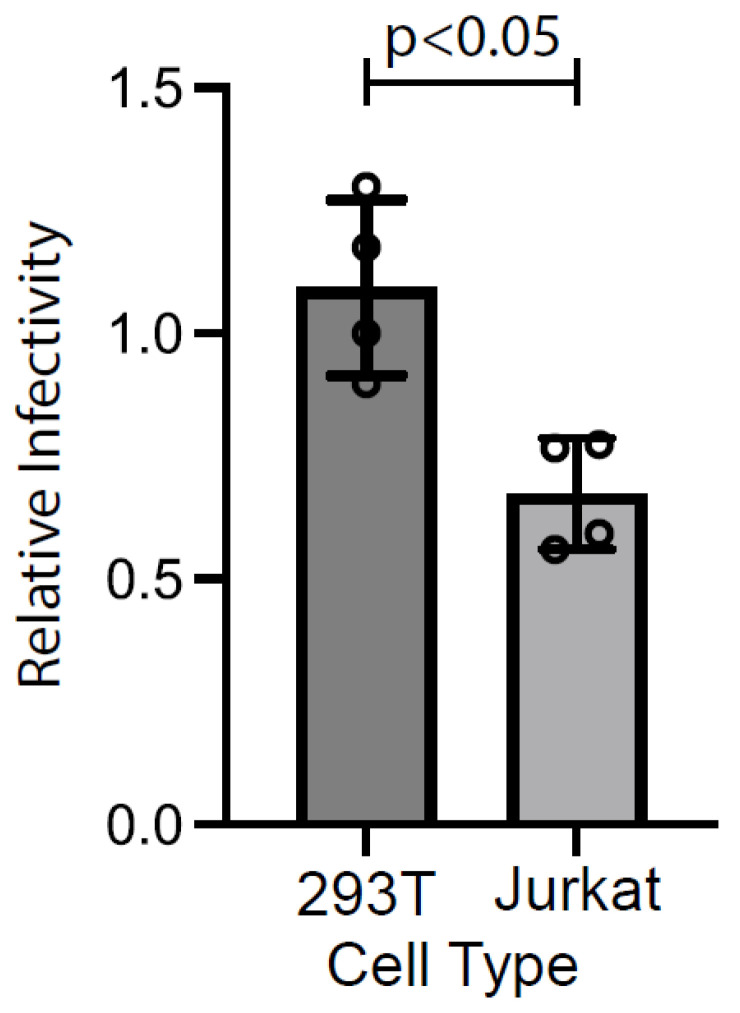
Comparing virus infectivity between HEK293T and Jurkat cells. Single-round infectivity assays using TZM-bl cells were performed to evaluate the differences in virus infectivity produced by HEK293T and Jurkat cells. The bar plot illustrates the relative infectivity of the FRET-labeled virus produced by Jurkat cells in comparison to its infectivity when produced by HEK293T cells. Error bars represent the standard deviation from four independent experiments. Statistical significance was determined using a Mann–Whitney U test.

## Data Availability

Data is contained within the article.

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
