# Peer review of "Evaluating HIV-1 Infectivity and Virion Maturation across Varied Producer Cells with a Novel FRET-Based Detection and Quantification Assay"

_ijms, 2024, doi:10.3390/ijms25126396_

Round 1

Reviewer 1 Report

Comments and Suggestions for Authors

The manuscript presents a study comparing the maturation rate and infectivity of HIV-1 virions in HEK293T and Jurkat cells. While the results are intriguing to certain readers, the perspective of the manuscript seems somewhat superficial and incomplete in explaining the underlying reasons for the observed differences between the two cell lines. As a result, the manuscript primarily highlights the phenomenon without delving into the underlying mechanisms. To improve the depth and comprehensiveness of the study, I recommend that the authors include additional experiments to enrich their analysis and provide insight into the mechanisms behind the observed differences.

Author Response

Reviewer’s comment: The manuscript presents a study comparing the maturation rate and infectivity of HIV-1 virions in HEK293T and Jurkat cells. While the results are intriguing to certain readers, the perspective of the manuscript seems somewhat superficial and incomplete in explaining the underlying reasons for the observed differences between the two cell lines. As a result, the manuscript primarily highlights the phenomenon without delving into the underlying mechanisms. To improve the depth and comprehensiveness of the study, I recommend that the authors include additional experiments to enrich their analysis and provide insight into the mechanisms behind the observed differences.

Authors’ Response: Thank you so much for reviewing our manuscript. We understand the importance, as suggested by the reviewer, of including more molecular mechanisms to explain why the virion maturation rate from the T-cell line is lower than that from HEK293 T-cells, which corresponds to the subsequent virus infectivity. As we discussed in the Discussion section, there are several factors contributing to this difference. One particular hypothesis is the expression differences of IP6, which has been reported to be important for virion maturation. Detecting IP6 is very challenging because it is a metabolite, making it unavailable for detection by western blot assay. However, we initiated a mass spectrometry analysis in collaboration with Dr. Monika Konaklieva, an associate professor in the Department of Chemistry at AU. We have identified the peak of IP6, confirming that we can measure its amount. We are currently developing a method for high-purity virus sample preparation for subsequent mass spectrometry analysis. This experiment will confirm our hypothesis about the role of IP6 in the observed phenomenon, though it is not the ultimate determinant. This manuscript is submitted in the communication format, which is intended for short articles presenting groundbreaking preliminary results or significant findings that are part of a larger, multi-year study. We believe that the observation of phenotypic differences in virus-producer cells meets the requirements of this format policy. We will conduct a deeper molecular biological analysis in the next phase of our research, which will be submitted as original research articles in the future. The requirement for further experiments and its difficulty have been emphasized in lines 133-135 in the discussion section highlighted in red.

Reviewer 2 Report

Comments and Suggestions for Authors

The article “Evaluating HIV-1 Infectivity and Virion Maturation Across Varied Producer Cells with a Novel FRET-Based Detection and Quantification Assay” differences in HIV-1 virion maturation and infectivity between viruses produced in HEK293T cells and Jurkat T cells. A FRET-based microscopy technique the authors had previously developed was used to quantify the maturation state of individual virions. A few suggestions to further improve the review:

Major comments:

1.      The introduction could provide more background on the different steps of the HIV viral entry process and where other inhibitors act such as entry inhibitors. This would help readers unfamiliar with HIV entry better understand the context for these drug classes a part from capsid inhibitors.

2.      A part form the FRET-based assay, additional validation using complementary techniques (e.g., electron microscopy, biochemical assays) is needed.

3.      The proposed mechanisms for the observed differences are plausible but remain speculative. Additional experiments targeting specific factors (e.g., IP6 levels, antiviral factors) could provide more direct evidence for the underlying mechanisms.

4.      Additional details on the Materials and Methods section are missing, such as the exact multiplicity of infection (MOI) used for virus production and infectivity assay.

Minor comments:

1.      In line 6: “Taisuke Izumi 1 and 2*” The sentence is incomplete or wrong.

Author Response

Reviewer’s comment: The article “Evaluating HIV-1 Infectivity and Virion Maturation Across Varied Producer Cells with a Novel FRET-Based Detection and Quantification Assay” differences in HIV-1 virion maturation and infectivity between viruses produced in HEK293T cells and Jurkat T cells. A FRET-based microscopy technique the authors had previously developed was used to quantify the maturation state of individual virions. A few suggestions to further improve the review:

Author’s Response: Thank you so much for taking the time to review our manuscript and for your valuable feedback. We have responded to each of your comments below.

Major comments:

Reviewer’s comment: The introduction could provide more background on the different steps of the HIV viral entry process and where other inhibitors act such as entry inhibitors. This would help readers unfamiliar with HIV entry better understand the context for these drug classes a part from capsid inhibitors.

Authors’ Response: According to the reviewer’s valuable suggestions, we added several sentences from lines 36 to 48 to provide more details about the HIV life cycle, including information on other inhibitors. These modifications have been highlighted in red.

Reviewer’s comment: A part form the FRET-based assay, additional validation using complementary techniques (e.g., electron microscopy, biochemical assays) is needed.

Authors’ Response: Our FRET-based assay was published in a peer-reviewed article in 2021, where we confirmed the corresponding maturation ratio using a TEM assay. This paper has been referenced by two different scientists, indicating that our system is recognized as an established method for addressing HIV-1 maturation. Therefore, we believe it is unnecessary to additionally confirm the correspondence of the FRET-based assay system in this manuscript. To emphasize that the correspondence of our tool has been confirmed with the TEM assay in a previous publication, we added two sentences to the background on lines 55 and 57. These modifications have been highlighted in red.

Reviewer’s comment: The proposed mechanisms for the observed differences are plausible but remain speculative. Additional experiments targeting specific factors (e.g., IP6 levels, antiviral factors) could provide more direct evidence for the underlying mechanisms.

Authors’ Response: We understand the importance, as suggested by the reviewer, of including more molecular mechanisms to explain why the virion maturation rate from the T-cell line is lower than that from HEK293 T-cells, which corresponds to the subsequent virus infectivity.

As we discussed in the Discussion section and in response to Reviewer 1’s comment, there are several factors contributing to this difference. One particular hypothesis is the expression differences of IP6, which has been reported to be important for virion maturation. Detecting IP6 is very challenging because it is a metabolite, making it unavailable for detection by western blot assay. However, we initiated a mass spectrometry analysis in collaboration with Dr. Monika Konaklieva, an associate professor in the Department of Chemistry at AU. We have identified the peak of IP6, confirming that we can measure its amount. We are currently developing a method for high-purity virus sample preparation for subsequent mass spectrometry analysis. This experiment will confirm our hypothesis about the role of IP6 in the observed phenomenon, though it is not the ultimate determinant. This manuscript is submitted in the communication format, which is intended for short articles presenting groundbreaking preliminary results or significant findings that are part of a larger, multi-year study. We believe that the observation of phenotypic differences in virus-producer cells meets the requirements of this format policy. We will conduct a deeper molecular biological analysis in the next phase of our research, which will be submitted as original research articles. The requirement for further experiments and its difficulty have been emphasized in lines 133-135 in the discussion section. These modifications have been highlighted in red.

Reviewer’s comment: Additional details on the Materials and Methods section are missing, such as the exact multiplicity of infection (MOI) used for virus production and infectivity assay.

Authors’ Response: Unfortunately, we did not determine the MOI for these viruses. Instead, we adjust the total injected number of viruses using the p24 amount measured by ELISA. We have previously determined that when we inject a dose-dependent number of viruses, the luciferase activity increases up to 100 ng of HIV-1 p24 in our assay. Therefore, 5 ng of HIV-1 p24 is not an oversaturated amount of virus particles. This method is consistent with our previous publication in Frontiers in Microbiology 2021, which has been well-established.

Minor comments:

Reviewer’s comment: In line 6: “Taisuke Izumi 1 and 2*” The sentence is incomplete or wrong.

Authors’ Response: Thank you for the suggestion. We have fixed this accordingly.

Round 2

Reviewer 1 Report

Comments and Suggestions for Authors

The issues have been addressed and the manuscript can be accepted.

Reviewer 2 Report

Comments and Suggestions for Authors

The authors have satisfactorily answered the comments and suggestions.